# Metformin inhibits mitochondrial complex I of cancer cells to reduce tumorigenesis

William W Wheaton[1†], Samuel E Weinberg[1†], Robert B Hamanaka[1], Saul Soberanes[1], Lucas B Sullivan[1], Elena Anso[1], Andrea Glasauer[1], Eric Dufour[2], Gokhan M Mutlu[1], GR Scott Budigner[1], Navdeep S Chandel[1*]

[1]Department of Medicine, The Feinberg School of Medicine, Northwestern University, Chicago, United States; [2]Institute of Biomedical Technology, University of Tampere, Tampere, Finland

**Abstract** Recent epidemiological and laboratory-based studies suggest that the anti-diabetic drug metformin prevents cancer progression. How metformin diminishes tumor growth is not fully understood. In this study, we report that in human cancer cells, metformin inhibits mitochondrial complex I (NADH dehydrogenase) activity and cellular respiration. Metformin inhibited cellular proliferation in the presence of glucose, but induced cell death upon glucose deprivation, indicating that cancer cells rely exclusively on glycolysis for survival in the presence of metformin. Metformin also reduced hypoxic activation of hypoxia-inducible factor 1 (HIF-1). All of these effects of metformin were reversed when the metformin-resistant *Saccharomyces cerevisiae* NADH dehydrogenase NDI1 was overexpressed. In vivo, the administration of metformin to mice inhibited the growth of control human cancer cells but not those expressing NDI1. Thus, we have demonstrated that metformin's inhibitory effects on cancer progression are cancer cell autonomous and depend on its ability to inhibit mitochondrial complex I.

*For correspondence: nav@ northwestern.edu

†These authors contributed equally to this work

Competing interests: The authors declare that no competing interests exist.

## Introduction

Metformin is widely used to treat patients with type II diabetes mellitus who have high levels of circulating insulin (*Nathan et al., 2009*). Metformin suppresses liver gluconeogenesis thereby reducing glucose release from the liver (*Inzucchi et al., 1998*; *Viollet et al., 2012*). In several recent retrospective studies, investigators have observed an association between metformin use and diminished tumor progression in patients suffering from different types of cancers (*Evans et al., 2005*; *Bowker et al., 2006*; *Dowling et al., 2012*). These data have prompted several prospective clinical trials to determine the efficacy of metformin as an anti-cancer agent. However, the underlying mechanism by which metformin diminishes tumor growth is not fully understood.

There are two postulated mechanisms by which metformin reduces tumor growth. Metformin may act at the organismal level, reducing levels of circulating insulin, a known mitogen for cancer cells. Alternatively, metformin may act in a cancer cell autonomous manner. Metformin is known to inhibit mitochondrial complex I in vitro (*Ota et al., 2009*; *El-Mir et al., 2000*; *Owen et al., 2000*) and it is thus possible that this targeting of the electron transport chain could inhibit tumor cell growth (*Birsoy et al., 2012*). This latter hypothesis has been questioned as cancer cells have the ability to survive on ATP produced exclusively by glycolysis. Furthermore, cancer cells have been shown to conduct glutamine-dependent reductive carboxylation to generate the TCA cycle intermediates required for cell proliferation when the electron transport chain is inhibited (*Mullen et al., 2012*; *Fendt et al., 2013*). Thus, it is not clear whether inhibition of complex I by metformin would result in decreasing tumor growth. In the present study, we directly tested whether inhibition of cancer cell mitochondrial complex I by metformin was required to decrease cell proliferation in vitro and tumor progression in vivo.

**eLife digest** Metformin is widely used to reduce the high blood sugar levels caused by diabetes. Recently, several studies have suggested that patients taking metformin who also develop cancer have tumors that grow more slowly than average. As clinical trials have already started to investigate if metformin is an effective anti-cancer treatment, it is important to understand how it might restrict tumor growth.

Researchers have proposed two ways that metformin could affect tumors. First, insulin is known to prompt cancer cells to divide, so the slower rate of tumor growth could just be a side-effect of the metformin reducing the amount of insulin in the blood. Alternatively, metformin could target cancer cells more directly by cutting the energy supply produced by their mitochondria. Metformin has been shown to disrupt complex I of the electron transport chain that is used by cells to generate energy. However, it is not known if disrupting complex I would actually stop cells dividing because they can generate energy in other ways.

Wheaton, Weinberg et al. have now demonstrated that metformin does target complex I in cancer cells, and that its effects depend on the amount of glucose available for cells to convert, without involving mitochondria, into energy. When there is plenty of glucose, metformin slows down the rate at which cancer cells divide, which slows down tumor growth. When the cells are deprived of glucose, metformin kills the cells instead.

Metformin also inhibits the pathways that regulate hypoxia inducible factors (HIFs), which are part of a system that helps cells to survive low-oxygen conditions, a prominent feature of many tumors. This means that metformin may combat cancer more effectively if used alongside other treatments that reduce the availability of both oxygen and glucose inside cells. Metformin could also potentially treat conditions that are linked to overactive HIFs, such as pulmonary hypertension.

## Results

Human HCT116 p53[−/−] colon cancer cells have previously shown to be sensitive to metformin (*Buzzai et al., 2007*). To determine if metformin treatment inhibited cellular oxygen consumption in these cells, we treated HCT116 p53[−/−] cells with increasing concentrations of metformin in media containing the metabolic substrates glucose, pyruvate, and glutamine for 24 hr. Subsequently, we measured cellular oxygen consumption. Metformin inhibited cellular oxygen consumption of HCT 116 p53[−/−] cells at concentrations (0.25–1.0 mM) similar to those reported to affect bioenergetics and gluconeogenesis in primary hepatocytes in vitro (*Figure 1A*; *Foretz et al., 2010*; *Miller et al., 2013*).

To determine whether metformin's inhibition of cellular oxygen consumption depended on mitochondrial complex I, we stably overexpressed the *Saccharomyces cerevisiae* protein NDI1 in HCT 116 p53[−/−] cells (hereon referred to as NDI1-HCT 116 p53[−/−] cells). NDI1 is a single-subunit NADH dehydrogenase, which oxidizes NADH in a process similar to the multi-subunit mammalian complex I; however without proton pumping or ROS generation (*Seo et al., 1998*). By contrast, mammalian complex I contains 45 subunits that pumps protons and generates ROS. NDI1-HCT 116 p53[−/−] cells demonstrated a slight, non-significant elevation in basal cellular oxygen consumption compared to control cells and were completely resistant to the effects of metformin on cellular oxygen consumption (*Figure 1—figure supplement 1*, *Figure 1B*).

To ensure that the inhibition of cellular oxygen consumption by metformin was a direct effect of metformin on complex I, we examined mitochondrial respiratory function in saponin-permeabilized cells. Saponin removes cholesterol from plasma membranes, allowing the entry of metabolic substrates directly to mitochondria (*Jamur and Oliver, 2010*). In the presence of ADP and the complex I substrates pyruvate and malate, metformin fully inhibited oxygen consumption in permeabilized Control-HCT 116 p53[−/−] cells (*Figure 1C*). By contrast, metformin had no effect on pyruvate/malate-driven oxygen consumption in NDI1-HCT 116 p53[−/−] cells (*Figure 1D*). Metformin also had no effect on oxygen consumption in saponin-permeabilized cells respiring on the complex II substrate succinate in the presence of ADP (*Figure 1E*). Interestingly, in saponin-permeabilized cells, metformin significantly inhibited complex I-dependent respiration at a much lower concentration than that required to inhibit oxygen consumption of intact cells, suggesting that transport across the plasma membrane is a barrier to metformin's inhibition of complex I. Metformin is known to slowly accumulate in cells in which its

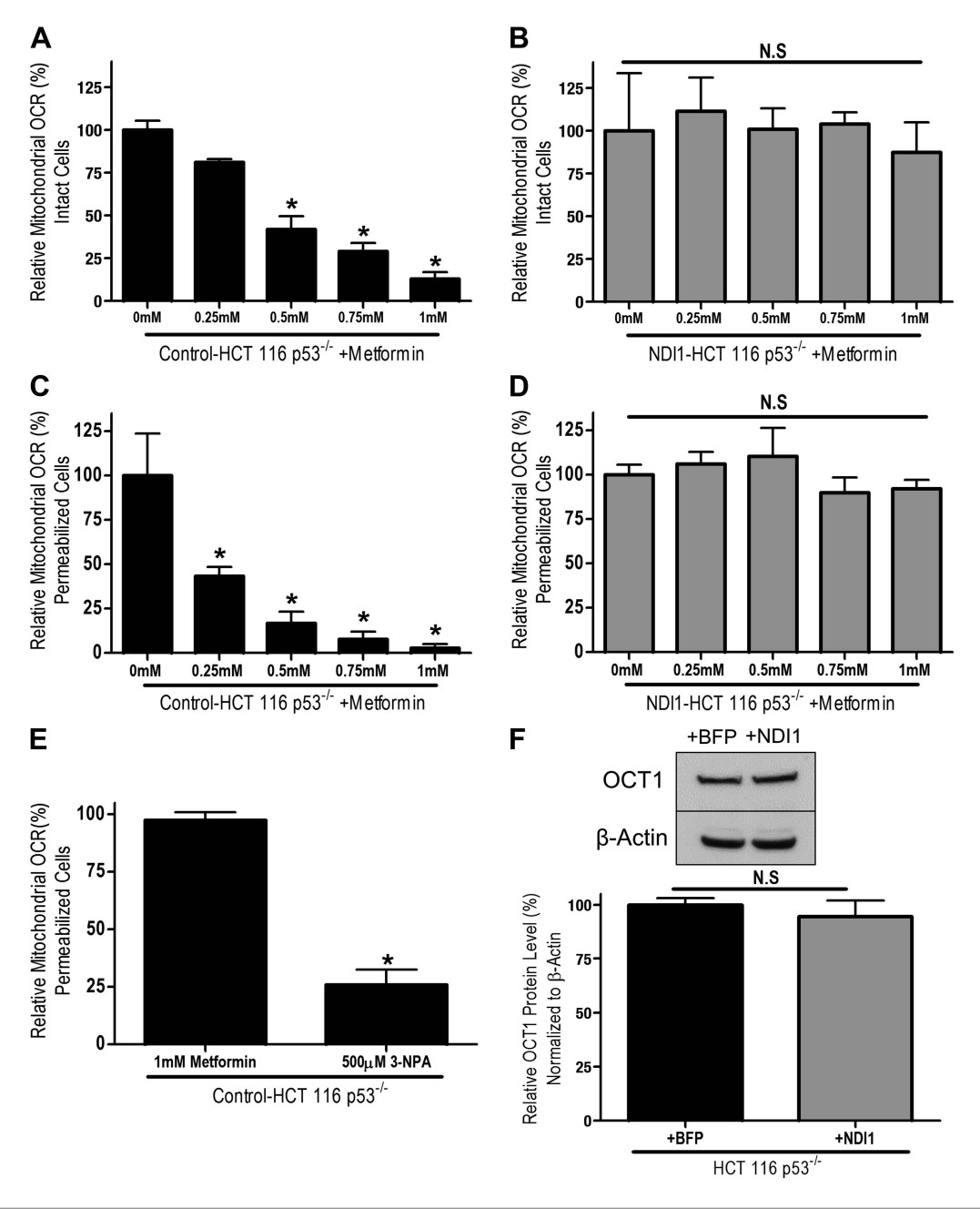

**Figure 1**. Metformin inhibits mitochondrial complex I function. (**A**) Relative mitochondrial oxygen consumption rate (OCR) of intact Control-HCT 116 p53$^{-/-}$ and (**B**) NDI1-HCT 116 p53$^{-/-}$ cells treated with metformin in complete media for 24 hr. (**C**) Relative complex I (2 mM malate, 10 mM pyruvate, 10 mM ADP)-driven oxygen consumption rate of saponin permeabilized Control-HCT 116 p53$^{-/-}$ cells and (**D**) NDI1-HCT 116 p53$^{-/-}$ cells treated with metformin for 20 min in mitochondrial assay buffer. (**E**) Relative complex II-driven oxygen consumption rate of saponin permeabilized Control-HCT 116 p53$^{-/-}$ cells treated with 10 mM succinate and 10 mM ADP in the presence of 1 mM metformin or the complex II inhibitor 3-Nitropropionic acid (3-NPA). (**F**) Representative western blot and quantification of levels of OCT1 protein in Control BFP-HCT 116 p53$^{-/-}$ and NDI1-HCT 116 p53$^{-/-}$ cells. Error bars are SEM (OCR: n = 4; OCT1: n = 4). * indicate significance p<0.05.

The following figure supplements are available for figure 1:

**Figure supplement 1**. NDI1 expression slightly increases oxygen consumption.

uptake is mediated by organic cation transporters (OCTs) (*Emami Riedmaier et al., 2013*). To ensure that NDI1-HCT 116 p53$^{-/-}$ cells are not refractory to metformin because of a change in metformin uptake, we analyzed the expression of OCT 1 in both control and NDI1-HCT 116 p53$^{-/-}$ cells. Expression of OCT1 protein did not change with the presence of NDI1 (*Figure 1F*).

We next sought to determine if metformin-dependent inhibition of complex I resulted in changes in proliferation and survival of HCT116 p53$^{-/-}$ cells. Metformin did not induce cell death in Control-HCT 116 p53$^{-/-}$ or NDI1-HCT 116 p53$^{-/-}$ cells in the presence of glucose (*Figure 2A,B*), however, in the absence of glucose, metformin induced cell death in Control-HCT 116 p53$^{-/-}$ but not in NDI1-HCT 116 p53$^{-/-}$ cells (*Figure 2C,D*). Metformin diminished cell proliferation in Control-HCT 116 p53$^{-/-}$ cells but not in NDI1-HCT 116 p53$^{-/-}$ cells in media containing glucose (*Figure 2E,F*).

These results indicate that the metformin-dependent inhibition of complex I decreases cell proliferation in the presence of glucose and increases cell death under glucose deprivation. These inhibitory effects of metformin were not specific to HCT116 p53$^{-/-}$ cells as metformin inhibited oxygen consumption and cellular proliferation of Control-HCT 116 p53$^{+/+}$ cells and Control-A549 human lung cancer cells but not NDI1-HCT 116 p53$^{+/+}$ or NDI1-A549 cells (*Figure 2—figure supplement 1 and 2*). Taken together, these results indicate that the anti-proliferative and cell death promoting effects of metformin require mitochondrial complex I inhibition.

We also examined whether phenformin, a more lipophilic biguanide, also exert its anti-proliferative effects on cancer cells through inhibition of complex I. Phenformin inhibited oxygen consumption in Control-HCT 116 p53$^{-/-}$ cells and saponin-permeabilized Control HCT 116 p53$^{-/-}$ cells at 100-fold lower concentration compared to metformin (*Figure 3A,C*). Expression of NDI1 rescued the phenformin-mediated decrease in oxygen consumption (*Figure 3B,D*). Phenformin diminished cell proliferation in the control but not NDI1 expressing HCT116 p53$^{-/-}$ cells (*Figure 3E,F*), and did not induce cell death in media containing glucose, similar to metformin (*Figure 3G,H*). Collectively, these results indicate that phenformin also exerts its biological effects through inhibition of mitochondrial complex I.

To determine whether metformin and phenformin diminish proliferation and survival of cells lacking endogenous complex I activity, we utilized a variant of CCL16 hamster fibroblasts that harbors a mutation in complex I (B2-CCL16) (*Seo et al., 1998*). Metformin inhibited proliferation of wild-type Control-CCL16 hamster fibroblasts but not that of B2-CCL16 cells or of B2-CCL16 cells reconstituted with NDI1 (NDI1-CCL16). Metformin and phenformin inhibited cellular oxygen consumption in wild-type CCL16 but not in CCL16-NDI1 cells (*Figure 3—figure supplement 1 and 2*). When these cells were cultured in galactose-substituted media, both metformin and phenformin induced cell death only in wild-type CCL16 cells. Survival of NDI1-CCL16 cells was not affected by metformin or phenformin (*Figure 3—figure supplements 1 and 2*). The B2-CCL16 cells die in galactose in the absence of metformin or phenformin since they harbor a mutation in complex I. Taken together, these results confirm that the anti-proliferative effects of metformin and phenformin require mitochondrial complex I inhibition.

The positive charge of metformin has been proposed to account for it is accumulation within the matrix of mitochondria that exhibit a robust inner mitochondria membrane potential (*Owen et al., 2000*). Alternatively, the non-polar hydrocarbon-side chain of the drug could promote binding to complexes within mitochondrial membranes. We tested whether the mitochondrial membrane potential is necessary for metformin-dependent inhibition of complex I. Saponin-permeabilized Control-HCT 116 p53$^{-/-}$ cells were induced to respire on pyruvate/malate in the presence of either ADP or CCCP. Although both ADP and CCCP induce mitochondrial respiration, only CCCP depolarizes mitochondrial inner membrane potential. Metformin inhibited ADP but not CCCP stimulated oxygen consumption indicating that the metformin-mediated inhibition of mitochondrial complex I required polarized mitochondria (*Figure 4A,B*). Rotenone, an irreversible inhibitor of complex I, does not require polarized mitochondria to inhibit mitochondrial oxygen consumption (*Figure 4C,D*). Our results suggest that metformin would not be effective in suppressing complex I activity of intact cells if mitochondrial inner membrane potential was disrupted. Metformin-mediated inhibition of the electron transport chain diminishes proton pumping, which might depolarize the mitochondrial membrane, thus limiting accumulation of the drug. However, we did not observe a reduction in the mitochondrial inner membrane potential measured using TMRE fluorescent dye in Control and NDI1 HCT116 p53$^{-/-}$ cells after metformin treatment (*Figure 4E,F*). When electron transport function is inhibited, the ATP synthase can function in reverse such that it uses ATP generated by glycolysis to pump protons across the inner mitochondrial membrane, maintaining membrane potential (*Appleby et al., 1999*). The ATP synthase

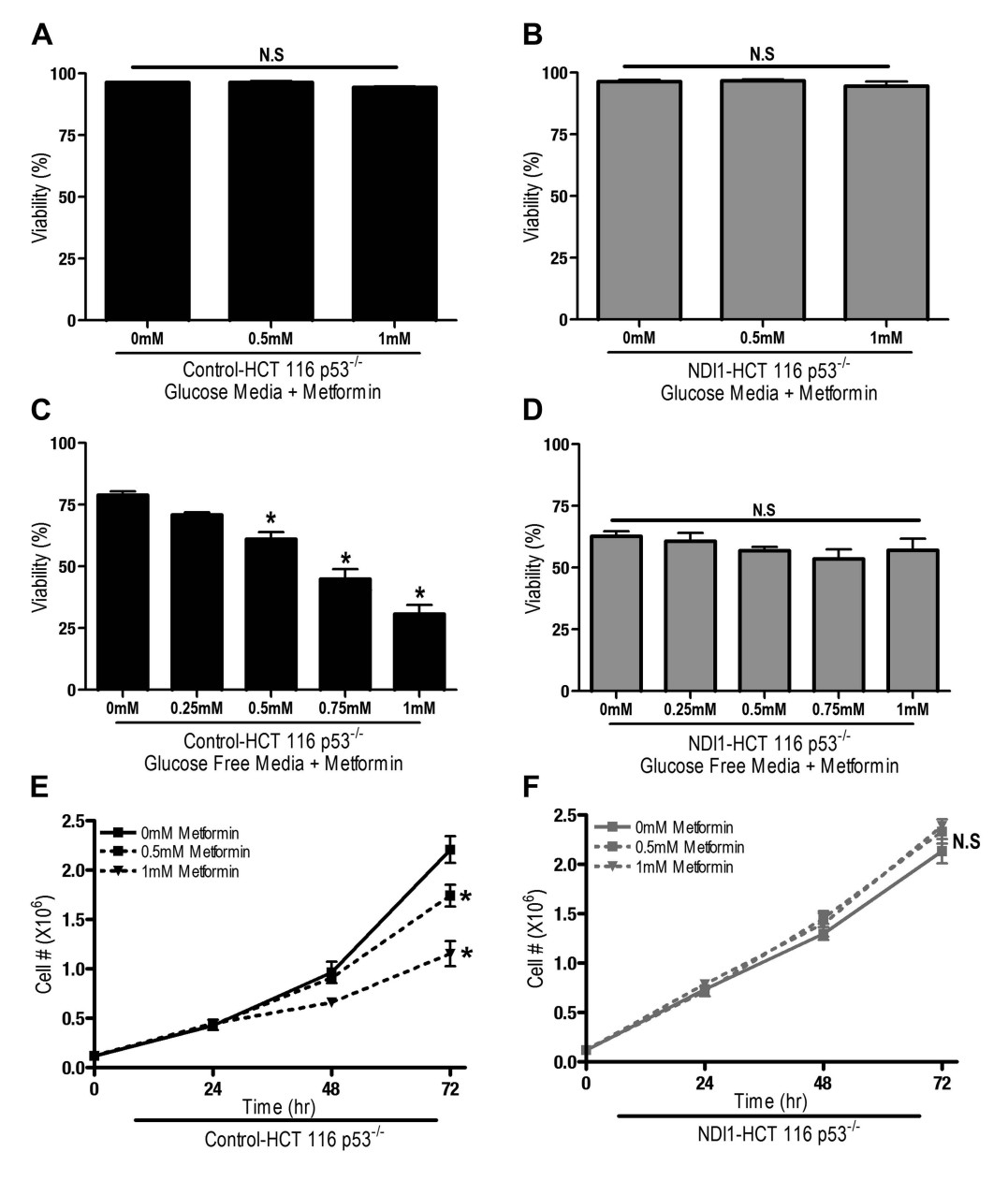

**Figure 2**. Metformin decreases cell proliferation by inhibiting mitochondrial complex I. (**A**) Percentage of live Control-HCT 116 p53$^{-/-}$ or (**B**) NDI1-HCT 116 p53$^{-/-}$ treated with metformin for 72 hr in media containing 10 mM glucose. (**C**) Percentage of live Control-HCT116 p53$^{-/-}$ or (**D**) NDI1-HCT 116 p53$^{-/-}$ treated with metformin for 24 hr followed by glucose withdrawal for 16 hr. (**E**) Cell number of Control-HCT 116 p53$^{-/-}$ cells and (**F**) NDI1-HCT 116 p53$^{-/-}$ cells 24, 48, and 72 hr post treatment with 0.5 mM or 1 mM metformin in complete media. Error bars are SEM (n = 4). * indicates significance p<0.05.

The following figure supplements are available for figure 2:

**Figure supplement 1**. Metformin decreases cellular proliferation through inhibition of mitochondrial complex I function in HCT 116 p53$^{+/+}$ cells.

**Figure supplement 2**. Metformin decreases cellular proliferation through inhibition of mitochondrial complex I function in A549 cells.

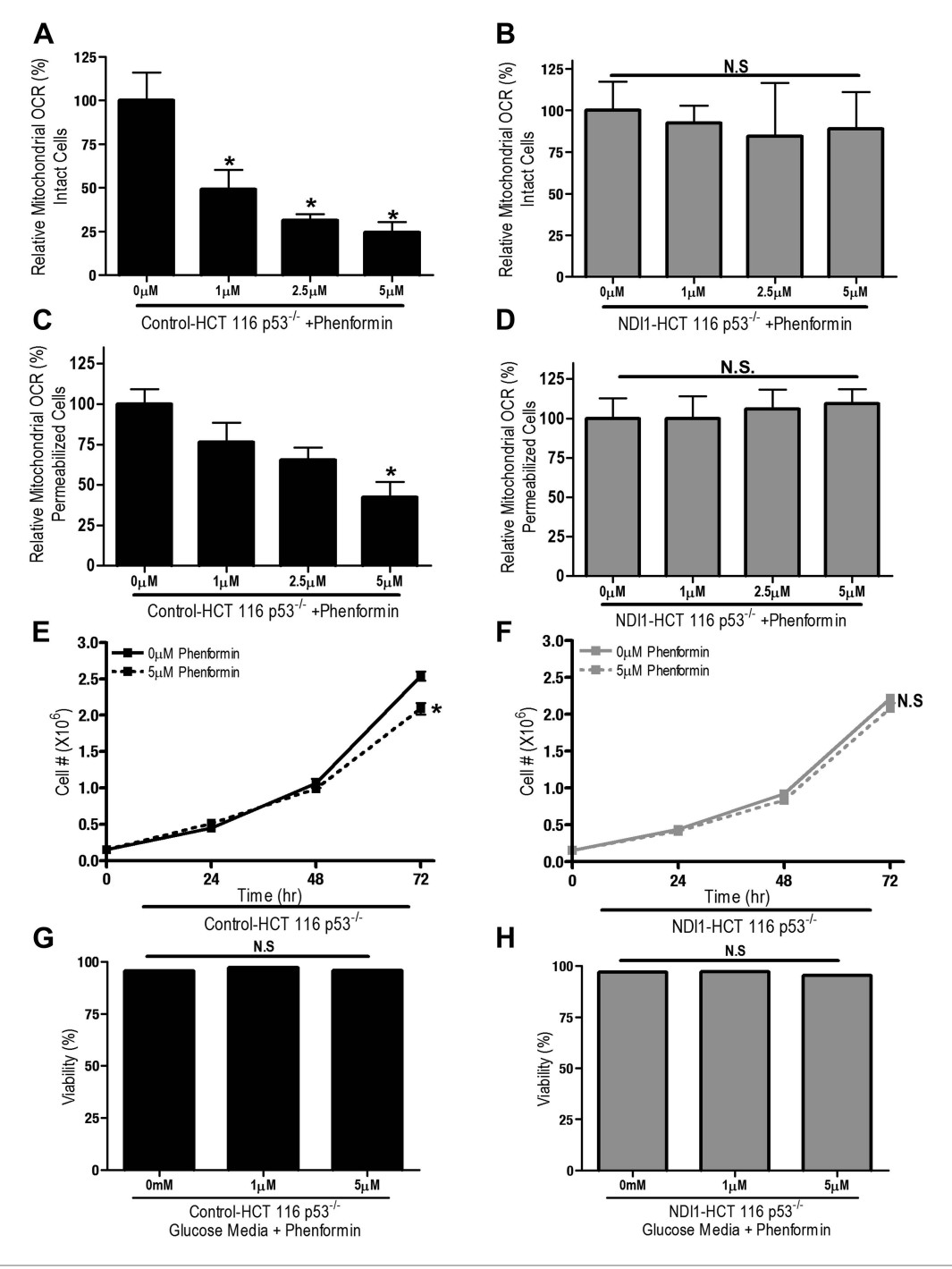

**Figure 3**. Phenformin decreases cell proliferation by inhibiting mitochondrial complex I. (**A**) Relative mitochondrial oxygen consumption rate (OCR) of intact Control-HCT 116 p53$^{-/-}$ and (**B**) NDI1-HCT 116 p53$^{-/-}$ cells treated with phenformin in complete media for 24 hr. (**C**) Relative complex I (2 mM malate, 10 mM pyruvate, 10 mM ADP)-driven oxygen consumption rate of saponin permeabilized Control-HCT 116 p53$^{-/-}$ cells and (**D**) NDI1-HCT 116 p53$^{-/-}$ cells treated with phenformin for 20 min in mitochondrial assay buffer. (**E**) Cell number of Control-HCT 116 p53$^{-/-}$ cells and (**F**) NDI1-HCT 116 p53$^{-/-}$ cells 24, 48, and 72 hr post treatment with 0 or 5 μM phenformin in complete media. (**G**) Percentage of live Control-HCT 116 p53$^{-/-}$ or (**H**) NDI1-HCT 116 p53$^{-/-}$ treated with metformin for 72 hr followed in complete media. Error bars are SEM (Relative OCR n = 5; Cell number n = 4). * indicates significance p<0.05.

*Figure 3. Continued on next page*

*Figure 3. Continued*

The following figure supplements are available for figure 3:

**Figure supplement 1**. Metformin inhibits mitochondrial complex I of CCL16 cells.

**Figure supplement 2**. Phenformin inhibits mitochondrial complex I of CCL16 cells.

---

inhibitor, Oligomycin A, diminished TMRE fluorescence in Control-HCT 116 p53$^{-/-}$ cells treated with metformin suggesting that in the presence of metformin, intact cells maintain their mitochondrial membrane potential by reversal of the ATP synthase (*Figure 4E*).

Rotenone irreversibly inhibits complex I, which contributes to its high toxicity in vivo. Because metformin is well tolerated, we sought to determine if metformin might reversibly bind to complex I. Saponin-permeabilized Control-HCT 116 p53$^{-/-}$ cells were treated with pyruvate and malate to maintain the mitochondrial inner membrane potential. Metformin was then added, followed by injection of either ADP or CCCP. Metformin inhibited ADP, but not CCCP-stimulated oxygen consumption (*Figure 5A,B*). As metformin accumulation requires mitochondrial membrane polarization (*Figure 4A,B*), these results indicate that metformin reversibly inhibits mitochondrial complex I. If the metformin that accumulated in the mitochondrial matrix irreversibly inhibited complex I, then oxygen consumption would have remained attenuated in CCCP-treated cells after metformin treatment.

An emerging function of mitochondria distinct from their ability to perform biosynthetic and bioenergetic reactions is the generation of H$_2$O$_2$, which promotes signaling in normal and cancer cells (*Hamanaka and Chandel, 2010*; *Sena and Chandel, 2012*). Mitochondrial complexes I and III produce superoxide into the mitochondrial matrix, where it is converted quickly to H$_2$O2 by SOD2 (*Brand, 2010*). Mitochondrial complex III also generates superoxide into the mitochondrial intermembrane space where it escapes through VDACs to cytosol and is converted into H$_2$O$_2$ by SOD1 (*Han et al., 2003*; *Muller et al., 2004*). We measured production and subsequent release of H$_2$O$_2$ from isolated mitochondria in the presence of metformin, rotenone, or antimycin A (complex III inhibitor) using pyruvate and malate as substrates.

Consistent with previous reports, rotenone and antimycin increased the release of H$_2$O$_2$ from mitochondria isolated from Control-HCT 116 p53$^{-/-}$ cells (*Figure 6A,B*; *St-Pierre et al., 2002*; *Muller et al., 2004*). In contrast, metformin did not substantially increase H$_2$O$_2$ release, suggesting that metformin and rotenone act on different sites of complex I (*Figure 6A*). When mitochondria were isolated from NDI1–HCT 116 p53$^{-/-}$ cells, only antimycin lead to a significant increase in H$_2$O$_2$ release (*Figure 6B*). Previous reports have shown that metformin does not substantially increase H$_2$O$_2$ production in isolated liver mitochondria and that metformin diminishes mitochondrial H$_2$O$_2$ production in response to paraquat, which induces mitochondrial ROS production (*Batandier et al., 2006*; *Algire et al., 2012*).

One biological consequence of mitochondrial-generated H$_2$O$_2$ is hypoxic stabilization of the hypoxia-inducible factors (HIFs) (*Chandel et al., 2000*). HIFs are involved in metabolic adaptation of tumor cells to hypoxia (*Semenza, 2012*). Metformin reduced hypoxic stabilization of HIF-1α in Control-HCT 116 p53$^{-/-}$ but not in NDI1-HCT 116 p53$^{-/-}$ (*Figure 6C,D*). Metformin did not decrease deferoxamine (DFO) stabilization of HIF-1α protein. DFO is an iron chelator known to directly stabilize HIF-1α protein independent of upstream signaling events. Metformin also significantly diminished hypoxic activation of HIF-dependent target genes, vascular endothelial growth factor (VEGF), and carbonic anhydrase 9 (CA9) in Control-HCT 116 p53$^{-/-}$ but not in NDI1-HCT 116 p53$^{-/-}$ (*Figure 6E*). Thus, metformin is an effective agent to reduce hypoxic activation of HIF-1.

Finally, we directly tested whether tumor cell autonomous inhibition of mitochondrial complex I by metformin was required to decrease tumor progression in vivo. As our NDI1-HCT 116 p53$^{-/-}$ cells are refractory to multiple effects of metformin in vitro, we reasoned that if metformin acted directly on mitochondrial complex I within the tumor cells to reduce tumorigenesis then NDI1-HCT 116 p53$^{-/-}$ xenograft tumors would not be inhibited in their growth. However, if metformin acts at the organismal level to diminish tumorigenesis then NDI1-HCT 116 p53$^{-/-}$ xenograft tumor growth would be suppressed similar to control tumors. Control-HCT 116 p53$^{-/-}$ cells subcutaneously injected into the left flank of nude mice rapidly grew in vivo, while tumors from mice fed metformin through drinking water ad libitum starting 4 days post-implantation exhibited a marked reduction in growth (*Figure 7A,B*). NDI1-HCT 116 p53$^{-/-}$ xenograft growth was resistant to metformin therapy (*Figure 7A,B*), suggesting

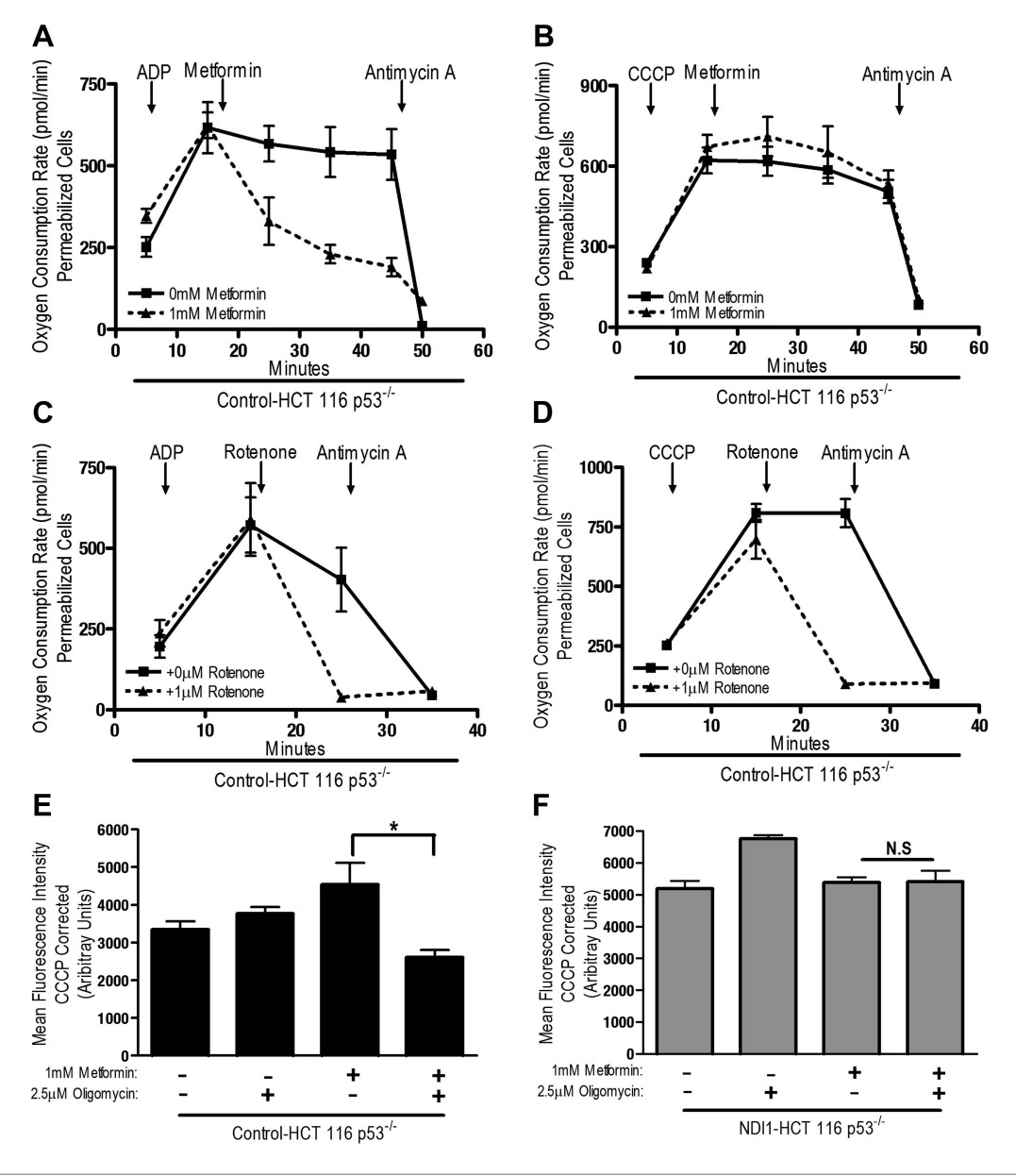

**Figure 4**. Metformin inhibition of complex I requires an intact mitochondrial inner membrane potential. (**A**) Complex I (2 mM malate, 10 mM pyruvate)-driven oxygen consumption rate of saponin permeabilized Control-HCT 116 p53$^{-/-}$ cells over time. At t = 5 min permeabilized cells were treated with either 10 mM ADP to induce respiration with an intact mitochondrial membrane potential or (**B**) 10 µM CCCP to induce respiration in absence of mitochondrial membrane potential. At t = 12 min 1 mM metformin was added to cells. At t = 48 min antimycin A was added. (**C**) Complex I (2 mM malate, 10 mM pyruvate)-driven oxygen consumption rate of saponin-permeabilized Control-HCT 116 p53$^{-/-}$ cells. At t = 5 min permeabilized cells were treated with either 10 mM ADP to induce respiration with an intact mitochondrial membrane potential or (**D**) 10 µM CCCP to induce respiration in absence of mitochondrial membrane potential. At t = 15 min, 1 µM rotenone was added to cells. At t = 25 min antimycin A was added. (**E**) Mitochondrial membrane potential measured by TMRE staining of Control-HCT116 p53$^{-/-}$ cells or (**F**) NDI1-HCT 116 p53$^{-/-}$ in the presence of 1 mM Metformin, 10 µM CCCP or 2.5 µM Oligomycin A. Error bars are SEM (n = 4). * indicates significance p<0.05.

that the metformin carries out its tumor inhibitory effects in a cancer cell autonomous manner through inhibition of mitochondrial complex I. Importantly, the consumption of water containing metformin was similar between control and NDI1 tumor barring mice (*Figure 7C*). Transcripts for the HIF target

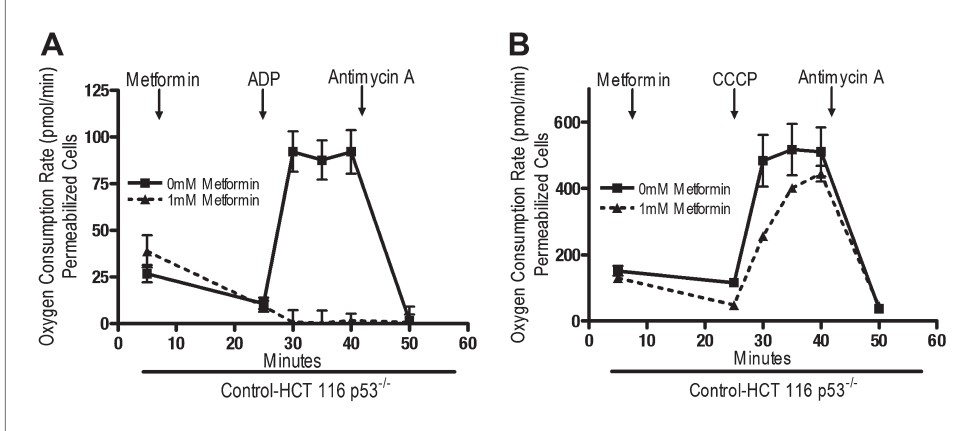

**Figure 5**. Metformin reversibly inhibits mitochondrial complex I. (**A**) Complex I (2 mM malate, 10 mM pyruvate)-driven oxygen consumption rate of saponin permeabilized Control-HCT 116 p53$^{-/-}$ cells over time. At t = 5 min permeabilized cells were exposed to 1 mM metformin. At t = 25 min respiration was stimulated with either 10 mM ADP to induce respiration with an intact mitochondrial membrane potential or (**B**) 10 μM CCCP to induce respiration lacking membrane potential with 10 mM ADP. At t = 42 min antimycin A was added. For mitochondrial membrane potential error bars are SEM (n = 4). For oxygen consumption rates, error bars are standard deviation (n = 6). * indicates significance p<0.05.

genes CA9 and VEGF were diminished in control tumors treated with metformin but not in NDI1 expressing tumors (*Figure 7D,E*). The blood glucose, plasma lactate, insulin, and IGF-1 level displayed no differences between the metformin-treated animals and control animals at the end of the study (*Figure 7—figure supplement 1*), consistent with previous reports (*Tomimoto et al., 2008*).

To further bolster our conclusions, we examined tumor growth of A549 cells expressing NDI1 and shRNA targeting mammalian NDUFS3, a subunit of the human mitochondrial complex I (*Vogel et al., 2007*). A549 cells are null for the tumor suppressor LKB1 and are known to be responsive to metformin therapy (*Rocha et al., 2011*). Furthermore, LKB1-deficient tumors are more susceptible to the related biguanide phenformin (*Shackelford et al., 2013*). We replaced the endogenous mitochondrial complex I by expressing the NDI1 protein in A549 cells stably expressing shRNA against NDUFS3 (*Figure 7—figure supplement 2A*), referred to as NDI1-shNDUFS3-A549 cells. The Control-A549 cells contain the empty vectors with selection markers BFP and puromycin for NDI1 and shRNA, respectively. NDI1-NDUFS3-A549 cells were resistant to the metformin-mediated reduction in cellular and mitochondrial oxygen consumption and cell proliferation (*Figure 7—figure supplement 2B–G*). Metformin also decreased HIF-1a protein levels in control but not in NDI1-NDUFS3-A549 cells (*Figure 7—figure supplement 3A*). Control-A549 cells subcutaneously injected into the left flank of nude mice rapidly grew in vivo, while tumors from mice fed metformin through drinking water *ad libitum* starting 2 weeks prior to tumor induction exhibited a marked reduction in growth over 45 days (*Figure 7—figure supplement 3B*). By contrast, NDI1-NDUFS3-A549 xenografts were completely resistant to metformin therapy, suggesting that metformin carries out its tumor inhibitory effects in a cell autonomous manner through inhibition of mitochondrial complex I in these cells (*Figure 7—figure supplement 3C*). NDI1-NDUFS3-A549 xenografts grew slower than control xenografts in untreated mice. An alternative explanation for the resistance of NDI1-NDUFS3-A549 cells to metformin could be that effects of metformin are blunted in slower-growing cells. However, based on our results from HCT116 cells in vivo and extensive analysis of A549 cells in vitro, we find that it is likely that NDI1 expressing A549 cells are also resistant to metformin in vivo due to rescue of complex I activity by the NDI1 protein.

## Discussion

The mechanisms by which metformin inhibits cancer growth is not fully understood. Metformin has been previously shown to inhibit mitochondrial complex I, yet it is not known whether metformin exhibits its anti-tumor effects through inhibition of complex I. It is important to note that many

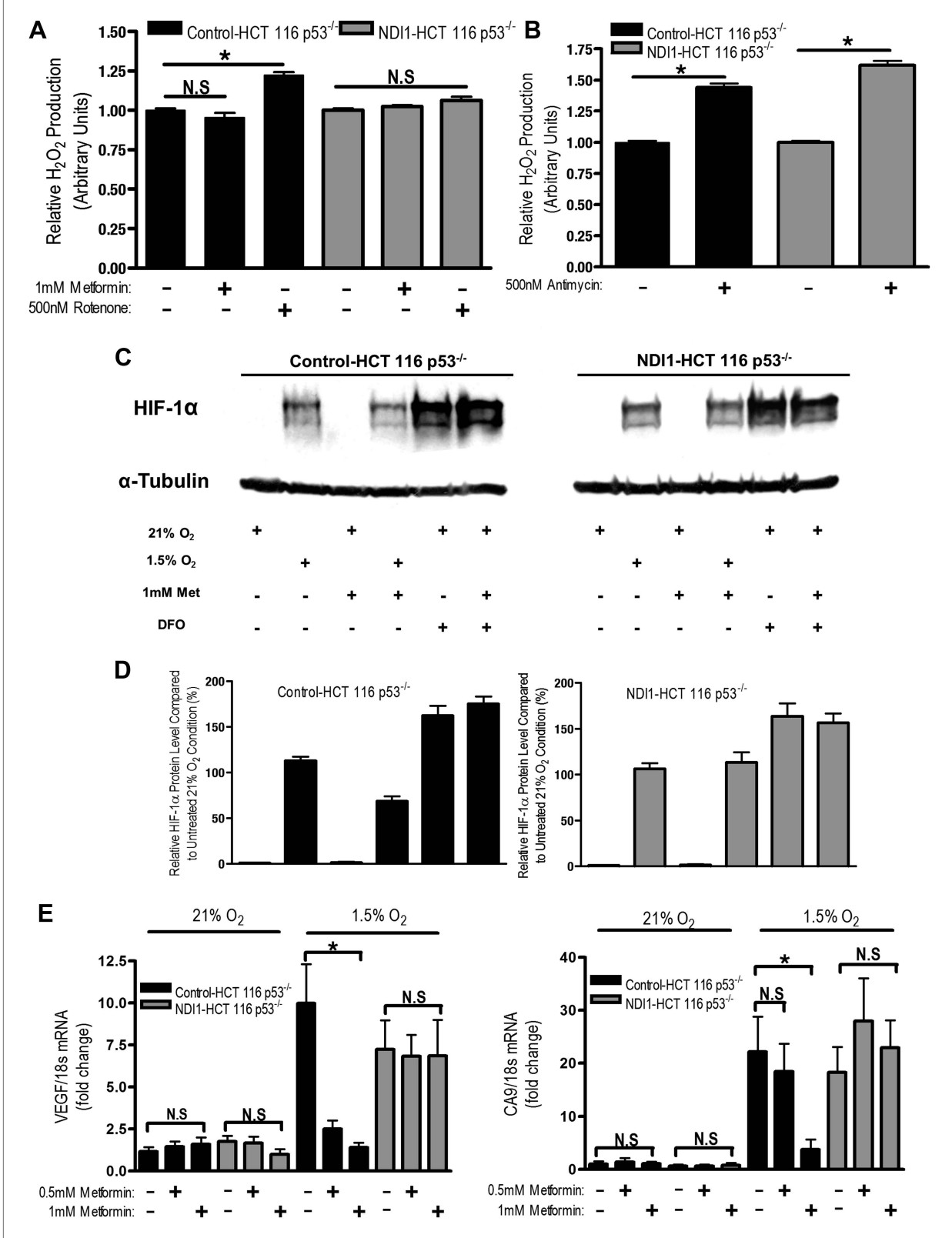

**Figure 6**. Metformin reduces HIF-1 activation through inhibition of mitochondrial complex I. (**A** and **B**) $H_2O_2$ levels emitted by mitochondria isolated from Control-HCT 116 p53$^{-/-}$ and NDI1-HCT 116 p53$^{-/-}$ cells respiring on 2 mM malate and 10 mM pyruvate. Mitochondria were treated with 1 mM Metformin, 500 nM rotenone, 500 nM Antimycin, or left untreated. $H_2O_2$ levels were measured using Amplex Red. (**C**) Levels of HIF1α protein in Control-HCT 116

*Figure 6. Continued on next page*

*Figure 6. Continued*

p53$^{-/-}$ and NDI1-HCT 116 p53$^{-/-}$ cells treated with 0 or 1 mM metformin for 24 hr, then placed in normoxia (21% O$_2$), hypoxia (1.5% O$_2$) or treated with Deferoxamine (DFO) for 8 hr. (**D**) Quantification of HIF1α protein levels from panel **C**. (**E**) Hypoxic-induced expression of HIF target genes in Control-HCT 116 p53$^{-/-}$ and NDI1-HCT 116 p53$^{-/-}$ treated with 0, 0.5 mM or 1 mM metformin for 24 hr following treatment with normoxia or hypoxia for 16 hr. Error bars are SEM (n = 3 for Amplex Red; Blot is representative of four independent blots quantified in **D**, n = 4 for gene expression). * indicates significance p<0.05.

commonly utilized drugs have been shown to inhibit mitochondrial function in vitro; however, for many of these drugs it is not clear whether the in vivo mechanism of action is through modulating mitochondrial metabolism (*Gohil et al., 2010*). In the present study, we genetically demonstrate that metformin acts as a complex I inhibitor in vitro and in vivo to inhibit tumorigenesis. Our current findings bolster our previous findings highlighting the importance of mitochondrial metabolism as essential for cancer proliferation (*Weinberg et al., 2010*; *Sullivan et al., 2013*). Our results are consistent with a recent report demonstrating another biguanide phenformin exerts its anti-tumor effects by inhibiting complex I (*Birsoy et al., 2014*). However, unlike metformin, phenformin has been discontinued because of frequent occurrence of lactic acidosis (*Bailey and Turner, 1996*). While our study highlights the importance of mitochondrial complex I inhibition within cancer cells as a major mechanism by which metformin reduces tumor burden, it does not necessarily preclude any additional organismal effects of metformin that might reduce tumor progression in certain cancers. Specifically, cancer cells that express insulin receptors might be affected by metformin's inhibition on hepatic gluconeogenesis to reduce circulating insulin levels (*Goodwin et al., 2012*).

The levels of metformin within cells is regulated by a balance between uptake mechanisms, which are dependent on expression of organic cation transporters (OCT 1, 2, and 3), and expulsion mechanisms, which are dependent on expression on multidrug and toxin extrusion proteins (MATE1 and MATE 2) (*Emami Riedmaier et al., 2013*). Cancer cells are likely to have a wide range in expression of uptake and extrusion proteins allowing for metformin accumulation. The uptake of metformin in mitochondria is also dependent on mitochondrial membrane potential. Metformin exists as a cation at physiological pH and thus its accumulation within mitochondria is predicted to increase as a function of the mitochondrial membrane potential. The inhibition of electron transport at complex I by metformin should reduce the mitochondrial membrane potential as proton pumping is linked to electron transport. However, we observed that metformin-treated cells maintain their mitochondrial membrane potential in part due to the reversal of the ATP synthase thus allowing accumulation of metformin in the mitochondrial matrix (*Figure 3*). It is well known that cells with compromised ETC can sustain mitochondrial membrane potential through this mechanism (*Appleby et al., 1999*). Collectively, our results indicate that will inhibit mitochondrial complex I and decrease tumorigenesis in cancer cells that have transporters across the plasma membrane, as well as a robust inner mitochondrial membrane potential to allow metformin to reach the mitochondrial matrix. It will be of interest to observe whether metformin's efficacy as an anti-cancer agent is dependent on the tumor expression of OCTs.

Our study also reveals that metformin's inhibition on mitochondrial complex I is distinct from the classical complex I inhibitor rotenone. Metformin requires a robust mitochondrial membrane potential to accumulate in the mitochondrial matrix and reversibly inhibits complex I. By contrast, rotenone is an irreversible inhibitor of complex I that does not require mitochondrial membrane potential. Rotenone accumulation is not dependent on specific transporters expressed in the plasma membrane and readily accumulates in all cells and thus is highly toxic. Furthermore, metformin does not promote generation of ROS from complex I, while rotenone stimulates ROS production from complex I (*Figure 5A*). The flavin site within Complex I produces ROS and previous studies indicate that rotenone inhibits complex I downstream of the flavin site, stimulating ROS generation (*Pryde and Hirst, 2011*; *St-Pierre et al., 2002*). It is likely that metformin acts upstream of this site, inhibiting complex I activity while also inhibiting ROS generation. Mitochondrial complex III also produces ROS (*Muller et al., 2004*). Therefore, metformin by inhibiting the complex I would limit electron flow to complex III thereby reducing ROS generation from complex III. A consequence of mitochondrial complex III ROS generation is the hypoxic activation of HIF-1 (*Bell et al., 2007*). Indeed, we found that metformin reduced hypoxic stabilization of HIF-1α protein and HIF-dependent target genes. This result is consistent with our previous study in which we identified multiple mitochondrial inhibitors in a large-scale chemical

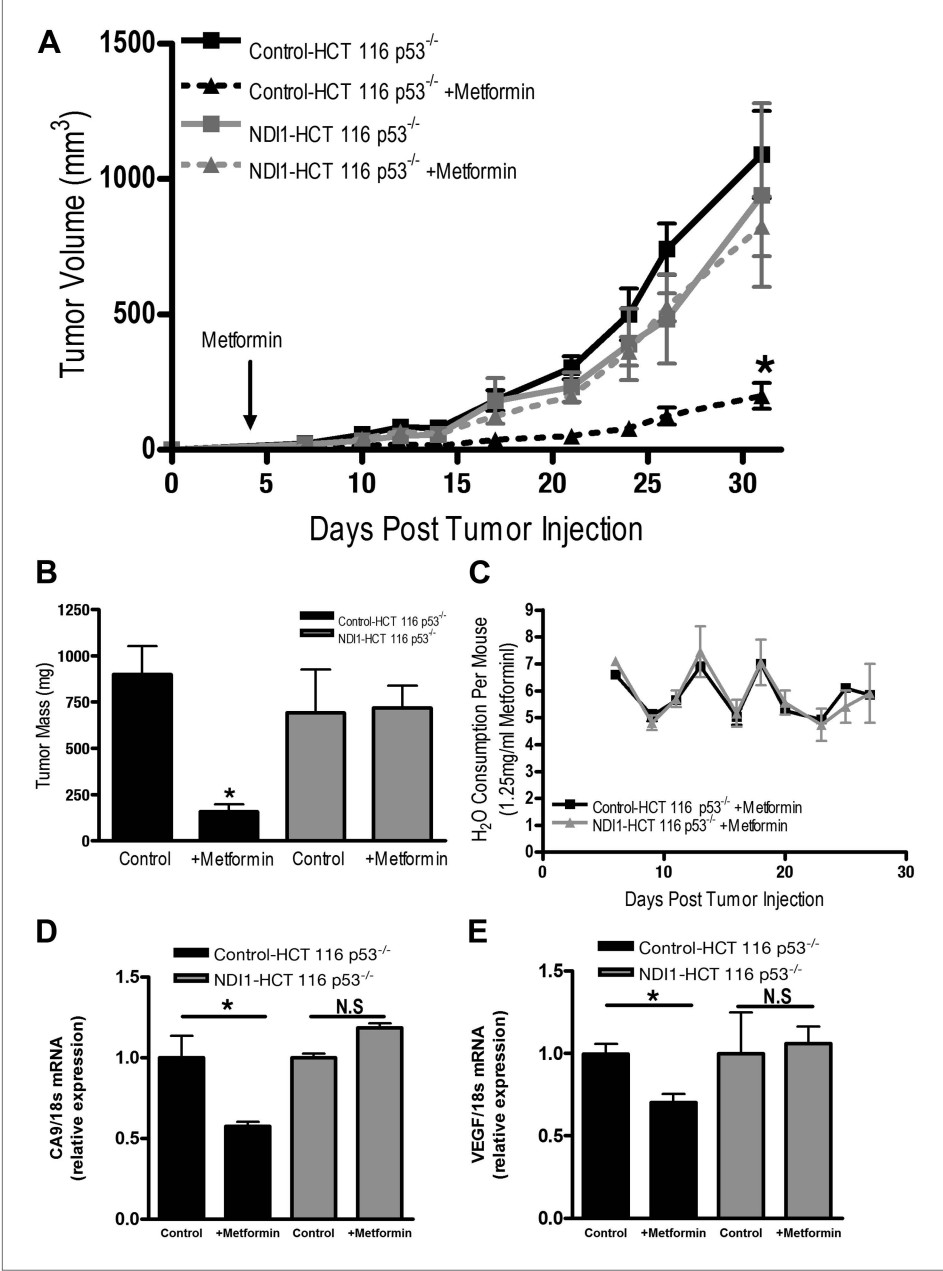

**Figure 7**. Metformin inhibits mitochondrial complex I to diminish tumor growth. (**A**) Average tumor volume in mice injected with $3 \times 10^6$ Control-HCT 116 p53$^{-/-}$ or NDI1-HCT 116 p53$^{-/-}$ cells injected into the left flank of J:Nu mice. Mice were given ad libitum, water free of metformin (squares) or were treated with 250 mg/kg of metformin in the drinking water starting 4 days post tumor injection (triangles). (**B**) Average tumor mass from mice injected with $3 \times 10^6$ Control-HCT 116 p53$^{-/-}$ or NDI1-HCT 116 p53$^{-/-}$ cells injected into the left flank of J:Nu mice after 32 days. (**C**) Average daily water consumption of mice treated with metformin (1.25 mg/ml). (**D**) HIF target genes expression measured in Control-HCT 116 p53$^{-/-}$ or NDI1-HCT 116 p53$^{-/-}$ tumors treated with metformin. Error bars are SEM (n = 8 per group for tumor study, n = 8 for $H_2O$ consumption, error bars represent standard deviation of two cages with four mice house in each cage, n = 3 for gene expression). * indicates significance p<0.05.

The following figure supplements are available for figure 7:

**Figure supplement 1**. Metformin treatment of tumor bearing mice does not alter blood glucose, plasma lactate, IGF-1, and insulin levels.

*Figure 7. Continued on next page*

*Figure 7. Continued*

**Figure supplement 2**. Metformin inhibits cellular proliferation and pro- proliferative signaling via complex I inhibition.

**Figure supplement 3**. NDI1 expressing A549 cells are refractory to metformin treatment in a xenograft model of tumor growth.

screen for inhibitors of hypoxic activation of HIF (*Lin et al., 2008*). Beyond cancer, metformin might be an effective treatment for diseases associated with hyperactivation of HIF such as pulmonary hypertension (*Shimoda and Semenza, 2011*). The concentration of metformin we administered to mice in this report is predicted to achieve plasma concentrations of metformin similar to those in humans receiving metformin therapy. In our study, metformin was administered to drinking water at a concentration of 1.25 mg/ml, approximately 250 mg/kg. Conversion of doses from mice to human is achieved using a formula based on body surface area normalization (*Reagan-Shaw et al., 2008*). The human equivalent dose of 250 mg/kg metformin in mice is 20.27 mg/kg or 1418 mg for a 70 kg adult. Patients typically receive 500–2500 mg metformin per day (*Scarpello and Howlett, 2008*). In these patients receiving metformin therapy, plasma levels range between 0.5–2 µg/ml; approximately 4 µM–15 µM (*Graham et al., 2011*). Previous studies have shown that mice receiving drinking water containing 1–5 mg/ml of metformin have a plasma steady-state metformin concentration of 0.45–1.7 µg/ml (*Memmott et al., 2010*), indicating that our metformin dosage falls within a clinically relevant range.

Metformin concentrations (250–1000 µM) used for in vitro experiments in this report are in the range utilized by investigators in the diabetes field to examine the effects of metformin on primary murine and human hepatocytes (*Foretz et al., 2010*; *Stephenne et al., 2011*; *Miller et al., 2013*). A perplexing observation revealed in our study and previous studies is that the metformin concentrations required to induce biological effects in vitro are a magnitude of order higher than plasma level concentrations of metformin in vivo. A likely cause for this difference is the accumulation of metformin in liver or tumors to local concentrations that are much higher than in the circulating plasma. Indeed, the gut has been shown to accumulate metformin in the mM range (*Bailey et al., 2008*; *Proctor et al., 2008*).

In summary, our results indicate that metformin reversibly inhibits mitochondrial complex I within cancer cells to reduce tumorigenesis. Metformin inhibits tumorigenesis through multiple mechanisms including the induction of cancer cell death in conditions, when glucose is limited and through inhibition of mitochondrial ROS-dependent signaling pathways that promote tumorigenesis (i.e., HIF). These results indicate that metformin would be most effective in low glucose and oxygen conditions. It will be of interest to determine whether metformin treatment might provide a useful adjunct to therapies that limit glucose uptake (e.g., PI3K inhibitors) or drive tumors to low glucose and oxygen levels (e.g., anti-angiogenic inhibitors).

## Materials and methods

### Generation of cell lines and culture

NDI1 (+NDI1) and control pWPI vectors containing BFP (+BFP) were transfected into 293FT cells using lipofectamine 2000 (Invitrogen, Carlsbad, CA, USA) along with pMD2.G and psPAX2 packaging vectors to produce Control-BFP or NDI-BFP lentivirus. Control-HCT116 p53$^{-/-}$, NDI1-HCT 116 p53$^{-/-}$, Control-A549, and NDI1-A549 were created by infecting parental HCT116 p53$^{-/-}$ or A549s with either Control-BFP lentivirus or NDI1-BFP lentivirus. Selection of the BFP expressing Control-HCT116 p53$^{-/-}$, NDI1-HCT 116 p53$^{-/-}$, Control-A549, and NDI1-A549 was done by periodic fluorescence-activated cell sorting (FACS) for BFP-positive cells with a MoFlo (Beckman–Coulter, Brea, CA, USA). Control-HCT 116 p53$^{-/-}$, NDI1-HCT 116 p53$^{-/-}$, Control-HCT116 p53$^{+/+}$, NDI1-HCT 116 p53$^{+/+}$, Control-A549, NDI1-A549 cells, NDI1-NDUFS3-A549, CCL16, CCL16-B2, and CCL16-NDI1 were cultured in DMEM supplemented with 10% fetal bovine serum, 1% HEPES, and 1% penicillin-streptomycin. Cells were routinely checked for BFP expression using FACS to ensure high NDI1 expression. TRC consortium validated pLKO.1 shRNA clones against control or NDUFS3 were obtained from Sigma and the lentivirus was produced in COS1 cells (TRCN0000218593). Control-A549 and NDI1-A549 cells infected the pLKO.1 lentivirus were additionally selected under continuous puromycin selection (1 µg/ml).

## Proliferation and cell viability

$1.5 \times 10^5$ HCT 116 wild-type or p53 null cells ($1 \times 10^5$ A549 and CCL16 cells) were plated on 35-mm dishes. At 24, 48, and 72 hr after plating, cells were trypsinized and counted using a Vi-Cell (Beckman–Coulter, Brea, CA, USA). Cell viability percentage was determined by trypan blue exclusion after 24 hr of treatment with metformin in DMEM with 10% dialyzed fetal bovine serum, 1% HEPES and 1% penicillin-streptomycin and lacking glucose (HCT 116) or with galactose substituted for glucose (CCL16).

## Oxygen consumption rate measurements

Oxygen consumption rates (OCR) were measured utilizing the XF24 Seahorse Biosciences Extracellular Flux Analyzer. $2 \times 10^4$ HCT 116 s ($1.5 \times 10^4$ A549 and CCL16 cells) were plated in the seahorse cell plate and incubated overnight. HCT 116 cells were then incubated for 24 hr in metformin hydrochloride (Sigma-Aldrich, St. Louis, CA, USA). 30 min before assay, the media were changed to 500 µl of fresh media containing metformin. For A549 and CCL16 cells, metformin was injected onto the cells using the Flux Analyzer and OCR was measured following a 20-min incubation. Basal mitochondrial oxygen consumption rate was determined by subtracting the antimycin A (Sigma, 1 µM) sensitive OCR from the basal OCR following normalization for cell number. The coupled mitochondrial oxygen consumption rate was determined by subtracting the mitochondrial respiration following Oligomycin A (Sigma) treatment from the basal mitochondrial oxygen consumption rate. The complex I and complex II specific contributions to oxygen consumption were measured by changing $5 \times 10^4$ HCT 116 cells ($2 \times 10^4$ A549 cells) plated overnight in complete DMEM into mitochondrial assay buffer (70 mM sucrose, 220 mM mannitol, 10 mM $KH_2PO_4$, 5 mM $MgCl_2$, 2 mM HEPES, 1.0 mM EGTA, and 0.2% (wt/vol) fatty-acid free BSA, pH 7.2), supplemented with 10 mM ADP (Sigma) and permeabilized by saponin injection (120 µg/ml, Sigma). The OCR was observed after saponin injection for loss of $O_2$ usage until stabilization at baseline where intracellular substrates have diffused out of the permeabilized cells. The complex I substrates (pyruvate 10 mM, malate 2 mM) or the complex II substrate (succinate 10 mM) was then added and the increase in OCR was measured. For complex II OCR, rotenone (Sigma, 1 µM) was added to inhibit complex I oxygen consumption. Post-injection of mitochondrial substrates, metformin was added and the OCR was measured. Complex I or II OCR was determined as the difference between substrate-free and substrate-added OCR. To examine whether mitochondrial inhibitors depend on membrane potential to inhibit complex I, cells were treated with saponin (120 µg/ml) and the complex I substrates (pyruvate 10 mM, malate 2 mM). Next, the cells were treated with either Carbonyl cyanide *m*-chlorophenyl hydrazone (CCCP, Sigma −10 µM) or ADP (10 mM) to induce respiration. The complex I inhibitors metformin (1 mM) or rotenone (1 µM) were then added to assay inhibitory capacity in the presence or absence of membrane potential. Finally, the cells were treated with antimycin (1 µM) to completely inhibit mitochondrial oxygen consumption. To determine the reversibility of metformin inhibition of complex I, cells were permeabilized with saponin and concurrently treated with complex I substrates (pyruvate 10 mM, malate 2 mM). The cells were then treated with 0 or 1 mM metformin for 20 min. Next, the cells were treated with CCCP (10 µM) or ADP (10 mM) to induce respiration in the presence or absence of mitochondrial membrane potential. Finally, the cells were treated with antimycin A (1 µM).

## Mitochondrial membrane potential measurements

$1 \times 10^6$ Control-HCT 116 p53$^{-/-}$ or NDI1-HCT 116 p53$^{-/-}$ cells were plated in 35-mm dishes in complete media and incubated overnight. The next day the cells were switched to media containing metformin and incubated for an additional 24 hr. After 24 hr, 50 nM tetramethylrhodamine (TMRE, Life Technologies) was added to the media for 30 min. The cells were washed, isolated, and resuspended in 1 ml of PBS. 10 µM CCCP, 2.5 µM Oligomycin A or no treatment was added and cells were incubated for an additional 30 min. The cells were then analyzed on BD LSRFortessa (San Jose, CA, USA) machine. The cells were gated for singlets and then the mean TMRE fluorescence was obtained using FlowJo analysis software.

## ROS measurements

The cells were scraped into mitochondrial isolation buffer (250 mM Sucrose, 1 mM EGTA, 20 mM Tris pH 7.4) and disrupted by 10 strokes with a Dounce homogenizer and 5 expulsions through a 28-gauge needle. The lysates were centrifuged at 500×$g$ for 10 min to remove nuclei, followed by centrifugation at 18,000×$g$ for 20 min to pellet mitochondria. Mitochondrial fractions were suspended in a reaction buffer containing 120 mM KCl, 5 mM $KH_2PO_4$, 3 mM HEPES, 1 mM EGTA, and 0.3% BSA, pH 7.2.

Mitochondria respired on 2.5 mM pyruvate and 1 mM malate in the presence or absence of 500 nm Antimycin A or 1 mM Metformin. Superoxide dismutase (200 U/ml; Sigma) was used to convert mitochondria-produced superoxide to hydrogen peroxide ($H_2O_2$). In the presence of horseradish peroxidase (10 U/ml, Thermo Scientific), Amplex Red (100 µM, Molecular Probes) reacts with $H_2O_2$, producing the fluorescent oxidation product, resorufin. Fluorescence was measured at excitation 544 nm, emission 590 nm.

## Immunoblot analysis

Protein was extracted using cell lysis buffer (Cell Signaling, Danvers, MA, USA) plus PMSF (600 µM). Protein concentration was quantified using the BCA Protein Assay (Pierce, Rockford, IL, USA). Protein samples were resolved on SDS polyacrylamide gels (Bio-Rad) and subsequently transferred to nitrocellulose membranes by semi-dry transfer using the Trans-Blot Turbo (Bio-Rad). To determine levels of HIF1α (BD Transduction Laboratories, Clone 54/HIF-1a) protein levels, HCT 116 cells were pretreated with 1 mM metformin or left untreated for 24 hr followed by treatment with 100 µM DFO, or incubation in 1.5% $O_2$ for 8 hr. For A549s, cells were pretreated with 2 mM metformin or mock treatment for 1 hr and treated with mock treatment, 100 µM DFO, or incubated in a hypoxia chamber at 1.5% $O_2$ for 4 hr. OCT1 (Novus Biologicals, Littleton, CO, USA) and NDUFS3 (MitoSciences) protein expression were assessed using the same technique above without any treatment conditions. α-tubulin (Sigma) was used as a loading control for protein blots assessing HIF1a and NDUFS3 protein levels while β-actin (Sigma) was used as a control for OCT1 protein quantification. Image Studio Lite version 3.1 (Licor) was used for analysis and quantification of protein levels. In brief, blots were scanned at high resolution and imported into Image Studio. Individual lanes of HIF1α and OCT1 protein levels were normalized to tubulin and actin respectively. Relative levels of HIF1a were normalized to an untreated sample incubated at 21% $O_2$.

## Real time PCR of HIF target genes

$1.5 \times 10^5$ Control-HCT 116 p53$^{-/-}$ or NDI1-HCT 116 p53$^{-/-}$ cells were plated on 24-well plates in complete media and 24 hr later treated with media without serum and metformin for 24 hr. Fresh media with or without metformin were added just before cells were exposed to either normoxia (21% $O_2$) or hypoxia (1.5% $O_2$) for 16 hr. Cells were washed once with ice-cold PBS and RNA was extracted using commercial kit (Aurum Total RNA mini Kit, Bio-Rad Hercules, CA). 1 µg of RNA was transcribed to cDNA using the iScript cDNA synthesis kit (Bio-Rad) and the relative concentrations of cDNA were analyzed by qPCR on a Bio-Rad CFX384 Touch Real-Time PCR Detection System using iQ SYBR Green Supermix (Bio-Rad) and the following primer sequences: CA9 sense: *CCGAGCGACGCAGCCTTTGA* CA9 antisense: *GGCTCCAGTCTCGGCTACCT* VEGF sense: *TACCTCCACCATGCCAAGTG* VEGF antisense: *GATGATTCTGCCCTCCTCCTT* 18s sense: *CGTTGATTAAGTCCCTGC CCTT* 18s antisense: *TCAAGTTCGACCGTCTTCTCAG.* HIF target gene expression was analyzed from tumor RNA isolated using TRIzol (Life Technologies, Grand Island, NY, USA). RNA was isolated and immediately transcribed to cDNA using the iScript cDNA synthesis kit, and the relative concentrations of cDNA were analyzed as described above.

## Blood glucose levels and ELISA assays

Blood was extracted weekly from female outbread athymic nude mice (J:Nu, Jackson Labs). The mice were injected with $1 \times 10^6$ A549 cells and treated with either water free of metformin or water containing 250 mg/kg metformin. Blood glucose levels were determined weekly using a glucometer (One Touch). Blood was collected into heparinized tubes at the end of the xenographic tumor study described below. Plasma was then isolated from whole blood by spinning at $21,000 \times g$ for 5 min. IGF-1 levels were determined using an ELISA (Abcam, Cambridge, MA, USA), as were insulin levels (Millipore, Billerica, MA, USA). Finally, lactate levels were measured using a colorimetric kit (Abcam).

## Mouse xenograft studies

Female J:Nu mice were injected with $3 \times 10^6$ Control-HCT 116 p53$^{-/-}$ or NDI1-HCT 116 p53$^{-/-}$ using a 27-gauge needle. 4 days post-injection mice were treated with water supplemented with metformin (Sigma) or water alone and fluid intake was monitored daily to ensure an effective dose of 1.25 mg/ml or 250 mg/kg (*Buzzai et al., 2007*) of metformin. In mice receiving drinking water containing 1–5 mg/ml of metformin, the plasma steady-state metformin concentration is reported to be in the range of 0.45–1.7 µg/ml (*Memmott et al., 2010*). In patients receiving metformin therapy, plasma levels are

between 0.5 µg/ml and 2 µg/ml (*Graham et al., 2011*). This treatment created four experimental groups: Control-HCT 116 p53$^{-/-}$ with $H_2O$ (n = 8), Control-HCT 116 p53$^{-/-}$ with Metformin (n = 8), NDI1-HCT 116 p53$^{-/-}$ with $H_2O$ (n = 8), NDI1-HCT 116 p53$^{-/-}$ with Metformin (n = 8). Experiments are from two independent cohorts of four mice each. Tumors were measured three times per week using calipers and tumor volume was determined using the equation ($3.14/6 \times L \times W^2$). At the completion of the study, mice were euthanized and the tumors were extracted and weighed. For A549s, $3 \times 10^6$ Control-A549 or NDI1-NDUFS3-A549 cells were injected into the left flank of female nude mice (Nu:Nu) administered metformin (250 mg/kg) in their drinking water for 2 weeks prior to cell injection and continuously administered throughout the experiment. All mouse work was done in accordance with Northwestern University Institutional Animal Care and Use Committee.

## Statistical analysis

Data are presented as the mean ± SEM. Statistical significance was determined using 1-way ANOVA with a Bonferroni posttest correction, 2-way ANOVA when two variables were present, or the students *t* test comparing control to experimental conditions for $p < 0.05$. For all differences uncovered, a student *t* test was performed to verify differences between control and experimental groups.

## Acknowledgements

We are grateful to the following people for providing reagents: E Scheffler (CCL16-B2 cells), Dr Takao Yagi (CCL16-NDI1 cells). This work is supported by grants from the NIH (R01CA123067, 5P01HL071643) to NSC, NIH (ES015024) to GMM and Veterans Administration Merit Award (GRSB). The work was also supported by NIH training grant T32 GM08061 to LBS and 5T32HL076139-10 to SW.

## Additional information

### Funding

| Funder | Grant reference number | Author |
| --- | --- | --- |
| National Institutes of Health | RO1 CA123067 | Navdeep S Chandel |
| National Institutes of Health | T32GM08061 | Lucas B Sullivan |
| National Institutes of Health | T32HL076139 | Samuel E Weinberg |

The funders had no role in study design, data collection and interpretation, or the decision to submit the work for publication.

### Author contributions

WWW, SEW, RBH, Conception and design, Acquisition of data, Analysis and interpretation of data, Drafting or revising the article; SS, LBS, EA, AG, Conception and design, Acquisition of data, Analysis and interpretation of data; ED, Conception and design, Contributed unpublished essential data or reagents; GMM, GRSB, NSC, Conception and design, Analysis and interpretation of data, Drafting or revising the article

### Ethics

Animal experimentation: Institutional animal approval: all mouse work was done in accordance with Northwestern University Institutional Animal Care and Use Committee approved protocol #2012-2840.

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
