## [Decision Letter]

Thank you for sending your work entitled “Metformin inhibits mitochondrial complex I of cancer cells to reduce tumorigenesis” for consideration at *eLife*. Your article has been favorably evaluated by a Senior Editor and 3 reviewers, one of whom, Chi Van Dang, is a member of our Board of Reviewing Editors.

The Reviewing editor and the other reviewers discussed their comments before reaching this decision, and the Reviewing editor has assembled the following comments to help you prepare a revised submission.

In the manuscript by Wheaton et al. the authors report experiments that support the hypothesis that postulates that metformin's biological effect in cancer biology is through inhibition of the mitochondrial Complex I NADH dehydrogenase. Metformin inhibited cellular oxygen consumption, reduced proliferation and induced death in the absence of glucose by targeting complex I of the mitochondrial respiratory chain in cancer cells. In addition, hypoxic activation of HIF-1 was reduced in metformin treated cells and metformin reduced the growth of tumour xenografts in mice. The experiments are well conceived and provide novel information related to the effects of metformin on metabolic and tumor growth control. The authors used a number of conventional approaches to pinpoint Complex I as a site of action of metformin, whose exact molecular function had been controversial in the literature. The authors used a single cell line HCT116, which lacks p53 to perform their experiments. Hence, whether the conclusions are generalizable could be equivocal. Nonetheless, the authors used the yeast equivalent of Complex I, the NDI1 NADH dehydrogenase, which is resistant to metformin, and showed it could rescue metformin inhibition of HCT116 cells. The authors also linked metformin mediated inhibition of HIF1 activity under hypoxia and demonstrated that this could also be rescued by NDI1. Lastly, NDI1 was able to rescue the inhibition of HCT116 tumorigenesis by metformin in vivo.

Overall, the authors provide a series of compelling experiments that support the contention that the target of metformin is Complex I NADH dehydrogenase. Short of direct structural (crystallography) data of a complex between metformin and mitochondrial Complex I, the studies by Wheaton et al. provide insight into the role of metformin but it is functional and not structural. There are a number of substantive concerns that should be addressed in the revised manuscript.

*A*. *Characterization of ND1 Overexpression*

1) These studies could be strengthened by the use of phenformin to demonstrate that the effects of a different chemical entity with activities like metformin could be blocked by the yeast NDI1. Extension beyond a single cell line would also strengthen the manuscript.

2) The property of NDI1 overexpression should be further characterized. Does expression of NDI1 change oxygen consumption?

3) What is the effect of metformin on cells with complex I knockdown? A complex I knockdown is needed to rule out a possible gain-of-function of NDI1.

4) Expression of OCT1 (organic cation transporter 1), the transporter required for uptake of metformin, should be examined in HCT116 and A549 cells, with or without NDI1, as the sensitivity of cancer cells to metformin may depend on expression of this transporter.

*B*. *Effects of Metformin on HIF*

1) Are the observed effects of metformin on cells in vitro also true in vivo, such as reduced proliferation, increased apoptosis and reduced glycolysis?

2) In the Results, it is stated that “metformin reduced hypoxic stabilization of HIF-1 in Control-HCT116 p53^-/-^ and Control-A549 cells but not in NDI1-HCT 116 p53^-/-^ and NDI1-A549 cells”. However, the levels of HIF-1 changes in the Western blots shown in Figure 5 are subtle to the eye. Additional Western blots and quantification of the images in Figure 5 should be provided.

3) The levels of HIF-1 should be assessed in the metformin treated xenograft tumours from Figure 6.

4) The amount of drinking water (with and without metformin) consumed during the tumour xenograft experiment depicted in Figure 6 should be reported. This will allow for an accurate estimate of the amount of metformin being administered to each mouse.

---

## [Author Response]

A. Characterization of ND1 Overexpression

*1) These studies could be strengthened by the use of phenformin to demonstrate that the effects of a different chemical entity with activities like metformin could be blocked by the yeast NDI1. Extension beyond a single cell line would also strengthen the manuscript*.

We have conducted experiments with phenformin and indeed the yeast NDI1 protein blocks phenformin-induced decrease in oxygen consumption and proliferation (Figure 3). Note that phenformin requires a log fold lower concentration than metformin to exert its effects. The revised paper includes 4 different cell lines where NDI1 is overexpressed. These include HCT116 p53^+/+^; HCT116 p53^-/-^; A549; CCL16-B2 cells. *E*

We also performed in vivo experiments using A549 cells that express the NDI1 protein and an shRNA targeting complex I subunit NDUSF3. This experimental setup allowed us to bolster our findings in HCT116 cells and test the effect of NDI1 expression in a human cell line which lacked endogenous complex I activity. One caveat to this experiment is that the tumor growth of the NDUSF3 shRNA/ND1 A549 cells is slower than their controls. Nevertheless, metformin substantially reduced control cells tumor growth while it had no effect on the growth of NDI1 expressing/NDUSF3 shRNA cells (Figure 7—figure supplement 3). We realize that an alternative explanation could be that metformin is not as effective in NDI1/NDUSF3 A549 cells due to their slower growth. We acknowledge this explanation in the current manuscript. Based on HCT116 cells and extensive in vitro data, it is likely that NDI1 expressing A549 cells are also resistant to metformin in vivo due to rescue of complex I activity by the NDI1 protein.

*2) The property of NDI1 overexpression should be further characterized. Does expression of NDI1 change oxygen consumption*?

We have conducted this experiment and demonstrated that NDI1 expression does not substantially change oxygen consumption (Figure 1—figure supplement 1).

*3) What is the effect of metformin on cells with complex I knockdown? A complex I knockdown is needed to rule out a possible gain-of-function of NDI1*.

To address this question, we used CCL16-B2 cells, which are complex I-deficient Chinese hamster cell mutant [Seo, B. B. et al. Molecular remedy of complex I defects: rotenone-insensitive internal NADHquinone oxidoreductase of Saccharomyces cerevisiae mitochondria restores the NADH oxidase activity of complex I-deficient mammalian cells. PNAS 95, 9167-9171 (1998)]. We now show that these complex I deficient cells are refractory to metformin and phenformin compared to their wild-type control cells. (Figure 3—figure supplement 1 and Figure 3—figure supplement 2). Furthermore, we performed in vivo experiments using A549 cells expressing an shRNA against the complex I structural protein NDUFS3. NDI1 expression in these cells did not contribute to a gain-of-function phenotype.

*4) Expression of OCT1 (organic cation transporter 1), the transporter required for uptake of metformin, should be examined in HCT116 and A549 cells, with or without NDI1, as the sensitivity of cancer cells to metformin may depend on expression of this transporter*.

This is an important experiment suggested by the reviewer. We have examined OCT1 protein levels between the control and NDI1 cell lines and the levels of OCT1 protein do not change (Figure 1).

B. Effects of Metformin on HIF

*1) Are the observed effects of metformin on cells in vitro also true in vivo, such as reduced proliferation, increased apoptosis and reduced glycolysis*?

Measurement of tumor glycolysis would be challenging for our lab since it requires glucose flux analysis by carbon labeling. We are not currently equipped to conduct this analysis in our lab. Metformin does not induce cell death in vitro unless glucose is limiting and this is a necrotic death that is unobservable by common apoptotic markers. We primarily examined HIF target genes in vivo since we felt this is novel finding in the paper (Figure 7). The antitumorigenic pathways invoked by complex I inhibition is beyond the scope of the current manuscript and will be focus of future studies.

*2) In the Results, it is stated that “metformin reduced hypoxic stabilization of HIF-1 in Control-HCT116 p53*^*-/-*^
*and Control-A549 cells but not in NDI1-HCT 116 p53*^*-/-*^
*and NDI1-A549 cells”. However, the levels of HIF-1 changes in the Western blots shown in*
Figure 5
*are subtle to the eye. Additional Western blots and quantification of the images in*
Figure 5
*should be provided*.

This experiment has been carried out in triplicate and we have now included

densitometric analysis of the blots (Figure 5).

*3) The levels of HIF-1 should be assessed in the metformin treated xenograft tumours from*
Figure 6.

We have measured the levels of HIF-dependent target gene expression in the xenograft tumors. Metformin decreases HIF target genes, CA9 and VEGF, in vivo (Figure 7).

*4) The amount of drinking water (with and without metformin) consumed during the tumour xenograft experiment depicted in*
Figure 6
*should be reported. This will allow for an accurate estimate of the amount of metformin being administered to each mouse*.

We now include the amount of drinking water consumed by mice during the course of our experiment (Figure 7).